# Soluble JAM-C Ectodomain Serves as the Niche for Adipose-Derived Stromal/Stem Cells

**DOI:** 10.3390/biomedicines9030278

**Published:** 2021-03-10

**Authors:** Morio Yamazaki, Kotaro Sugimoto, Yo Mabuchi, Rina Yamashita, Naoki Ichikawa-Tomikawa, Tetsuharu Kaneko, Chihiro Akazawa, Hiroshi Hasegawa, Tetsuya Imura, Hideki Chiba

**Affiliations:** 1Department of Basic Pathology, School of Medicine, Fukushima Medical University, Fukushima 960-1295, Japan; 0624m.yamazaki@gmail.com (M.Y.); m151124@fmu.ac.jp (R.Y.); naoichi2004@yahoo.co.jp (N.I.-T.); timura@koto.kpu-m.ac.jp (T.I.); 2Division of Dentistry and Oral Surgery, School of Medicine, Fukushima Medical University, Fukushima 960-1295, Japan; k-tetsu@fmu.ac.jp (T.K.); hi612hasegawa@gmail.com (H.H.); 3Department of Biochemistry and Biophysics, Graduate School of Medical and Dental Sciences, Tokyo Medical and Dental University, Tokyo 113-8510, Japan; yomabuchi.bb@tmd.ac.jp (Y.M.); c.akazawa.gt@juntendo.ac.jp (C.A.)

**Keywords:** mesenchymal stem cell, stem cell, niche, junctional adhesion molecule, tight junction, shedding

## Abstract

Junctional adhesion molecules (JAMs) are expressed in diverse types of stem and progenitor cells, but their physiological significance has yet to be established. Here, we report that JAMs exhibit a novel mode of interaction and biological activity in adipose-derived stromal/stem cells (ADSCs). Among the JAM family members, JAM-B and JAM-C were concentrated along the cell membranes of mouse ADSCs. JAM-C but not JAM-B was broadly distributed in the interstitial spaces of mouse adipose tissue. Interestingly, the JAM-C ectodomain was cleaved and secreted as a soluble form (sJAM-C) in vitro and in vivo, leading to deposition in the fat interstitial tissue. When ADSCs were grown in culture plates coated with sJAM-C, cell adhesion, cell proliferation and the expression of five mesenchymal stem cell markers, *Cd44*, *Cd105*, *Cd140a*, *Cd166* and *Sca-1*, were significantly elevated. Moreover, immunoprecipitation assay showed that sJAM-C formed a complex with JAM-B. Using CRISPR/Cas9-based genome editing, we also demonstrated that sJAM-C was coupled with JAM-B to stimulate ADSC adhesion and maintenance. Together, these findings provide insight into the unique function of sJAM-C in ADSCs. We propose that JAMs contribute not only to cell–cell adhesion, but also to cell–matrix adhesion, by excising their ectodomain and functioning as a niche-like microenvironment for stem and progenitor cells.

## 1. Introduction

Mesenchymal stem cells (MCSs) were first identified from the bone marrow [1] and are considered to reside in the perivasculature of a variety of organs [2,3,4]. MCSs show a capacity to self-renew and differentiate into multiple lineages and exhibit regenerative and immunoregulatory properties [4,5,6,7], thereby being under clinical trials for stem cell therapy against various diseases [8,9,10]. Among MSCs, adipose-derived stromal/stem cells (ADSCs) represent one of the most attractive tools for cellular therapy, because the fat tissue is easily accessible and ADSCs can be isolated even without selection using fluorescence-activated cell sorting (FACS).

Adult adipose tissues contain diverse types of cells. Mature adipocytes contain large lipid droplets and account for approximately 90% of the adipose tissues in volume, but their population is less than half of the total cell number [11]. The rest of the cell types are distributed in interstitial spaces between mature adipocytes and are called the stromal vascular fraction (SVF), which consists of endothelial cells, pericytes and smooth muscle cells of microvessels, as well as immune cells, fibroblasts and ADSCs [12]. The SVF-derived ADSCs are capable of differentiating into both adipocytes and microvascular components in vivo [12].

Junctional adhesion molecules (JAM-family proteins; JAMs) are single-spanning transmembrane proteins (30–45 kDa in range) that belong to the CTX (cortical thymocyte marker in *Xenopus*) subfamily in the immunoglobulin superfamily (IgSF) [13,14,15,16,17,18,19,20]. The JAM family includes JAM-A (*Jam1*, *F11r*), JAM-B (*VE-JAM*, *Jam2*), JAM-C (*Jam3*), JAM4 (*Igsf5*), JAML (*Amica1*), CAR (coxsackie virus and adenovirus receptor; *Cxadr*), CLMP (coxsackievirus and adenovirus receptor-like membrane protein; *Clmp*) and ESAM (endothelial cell-selective adhesion molecule; *Esam*). JAMs comprise two extracellular IgSF domains (a membrane-distal V-type and a membrane-proximal C2-type Ig domain), a single transmembrane region and a C-terminal cytoplasmic domain. They reveal homophilic and heterophilic interactions through their extracellular domains not only between the same types of cells but also among distinct types of cells [19,20,21]. On the other hand, the C-terminal cytoplasmic domain of JAMs is thought to propagate intracellular signals, since several signaling proteins interact with the C-terminus [19]. JAMs are expressed in diverse epithelial cells and in a range of nonepithelial cells such as endothelial cells, blood cells, myocytes, nervous system cells, Sertoli cells and spermatids, and contribute to numerous physiological and pathological events, e.g., cell adhesion, cell polarity, cell migration and cell death, as well as hematopoietic differentiation, spermatogenesis, neurogenesis and cancer progression.

JAMs are also detected in various types of stem and progenitor cells, including embryonic, neural and hematopoietic stem cells [13,14,15,16,17,18] as well as spermatogonia [19,20]. JAM-B and JAM-C are expressed on hematopoietic stem cells (HSCs) and bone marrow stromal cells, respectively, and their heterophilic interactions participate in homing hematopoietic stem cells to the bone marrow [21]. In addition, the JAM-A/JAM-B engagement between HSC precursor cells and somite stromal cells regulates HSC fate in zebrafish via the Notch signaling pathway [22]. However, the functional relevance of JAMs in stem and progenitor cells other than HSCs remains unknown.

In the present study, we uncovered the physiological significance of JAMs in ADSCs. We showed that both JAM-B and JAM-C are expressed on the cell surfaces of ADSCs. We found that the extracellular domain of JAM-C, but not of JAM-B, is cleaved and secreted as a soluble form (sJAM-C) from ADSCs, followed by deposition to the interstitium of the adipose tissue. Furthermore, we also demonstrated that sJAM-C is coupled with JAM-B to promote cell adhesion, cell growth and the expression of MSC markers in ADSCs. These findings highlight that JAM-C is involved not only in cell–cell adhesion but also in cell–matrix adhesion by shedding its ectodomain to act as a niche-like microenvironment for stem cells.

## 2. Materials and Methods

### 2.1. Antibodies

The antibodies used in this study are listed in Table 1.

### 2.2. Isolation and Culture of ADSC

For isolation of ADSCs, subcutaneous adipose depots dissected from 8-week-old male mice were minced and digested using 0.075% collagenase type I (SCR103, Sigma-Aldrich, St. Louis, MO, USA) for 45 min at 37 °C. The cell suspension was centrifuged at 200× *g* for 5 min to separate floating adipocytes from the SVF. After filtration of the suspension through a 70-µm cell strainer to remove cellular debris, the SVF cells were resuspended in Dulbecco’s modified Eagle’s medium (DMEM) containing 10% fetal bovine serum and were plated onto plastic culture dishes. Floating cells and debris were removed by refreshing the culture medium. The culture medium was changed every 2–3 days. After 7–10 days, the cells were stripped using 0.25% trypsin-EDTA solution and passaged into a new plate. To prepare the recombinant protein-coated plates, 1 µg of recombinant JAM-B (rJAM-B; 50464-M08H, SinoBiological, Beijing, China), recombinant JAM-C (rJAM-C; 50465-M08H, SinoBiological, Beijing, China), normal mouse IgG (12-371, Merck Millipore, Burlington, MA, USA) or Cellmatrix type I-A collagen (KP-2020, Nitta gelatin, Osaka, Japan) was placed in 12-well plates overnight at 4 °C. Since rJAM-B/C consists of two extracellular IgSF domains as described above, IgG was used as the negative control.

All animal experiments complied with the National Institutes of Health Guide for the Care and Use of Laboratory Animals and were approved by the Animal Committee at Fukushima Medical University (Approved number, 30060; date, 13 Apr 2018).

### 2.3. Genome Editing

For CRISPR vector construction, annealed DNA oligo-pairs complementary to mouse *Jam2* (caccgGCAGCATCAGGAGGCCTTGG and aaacCCAAGGCCTCCTGATGCTGCc) and *Jam3* (caccgATTCATGTACCACTGGGTTT and aaacAAACCCAGTGGTACATGAATc) exons were cloned into the lentiCRISPR v2 plasmid (#52961, Addgene) [23] at the *Esp3*I (*BsmB*I) site. HEK293T cells were transfected with 10 µg of the lentiCRISPR v2 plasmids, 5 µg of packaging plasmids psPAX2 (#12260, Addgene) and pCMV-VSV-G (#8454, Addgene) using PEI Max following a standard protocol. Culture media containing recombinant lentiviruses were collected 72 h after transfection and were directly added to ADSC with polybrene at 10 µg/mL. Then, 24 h after transfection, culture media were replaced by fresh DMEM with 1 µg/mL of puromycin. More than 7 days after transfection, the cells were used for further analysis.

### 2.4. RNA Extraction and RT-PCR

For analysis of gene expression, total RNA was isolated from ADSC using TRIzol RNA Isolation Reagents (15596018, Thermo Fisher Scientific, Waltham, MA, USA), and reverse transcription (RT) was performed using a SensiFAST™ cDNA Synthesis Kit (BIO-65053, Meridian Bioscience, Cincinnati, OH, USA). For PCR, the target sequences were amplified using GoTaq Green Master Mix (M7122, Promega, Madison, WI, USA). Aliquots of the PCR products were loaded onto 2.5% agarose gel and analyzed after staining with ethidium bromide. Original PCR values were quantified by ImageJ software ver 1.53c (Wayne Rasband National Institutes of Health). The expression levels of the target genes in RT-PCR were divided by the corresponding *Gapdh* signal intensity. Quantitative PCR (qPCR) was performed using the THUNDERBIRD SYBR qPCR Mix (QPS-201, TOYOBO, Tokyo, Japan) and Step One Real-Time PCR System ver. 2.0 (Applied Biosystems, Foster City, CA, USA). The primers for RT-PCR are listed in Table 2.

### 2.5. Immunoprecipitation and Western Blot

Total cell or tissue extracts were collected by using CellLytic^TM^ MT Cell Lysis Reagent (C3228, Sigma-Aldrich, St. Louis, MO, USA), and were subsequently sonicated with three or four bursts of 5–10 sec. For mouse adipose tissue, cell lysates were centrifuged for 5 min at 11,000× *g*, and the upper layer containing hydrophobic lipids was removed. For the immunoprecipitation assay, ADSCs were grown on a 27-mm glass-based dish (3960-035, IWAKI, Tokyo, Japan) coated with 50 µg of Cellmatrix type I-A collagen (KP-2020, Nitta gelatin, Osaka, Japan) to collect sJAM-C in the extracellular matrix. Immunoprecipitation was performed using an immunoprecipitation kit (Protein G; 11719386001, Sigma-Aldrich, St. Louis, MO, USA), following the manufacturer’s protocol.

Whole-cell lysates or the immunoprecipitated samples were mixed with sample loading buffer containing 2-mercaptoethanol and incubated for 10 min at 95 °C. They were resolved by one-dimensional SDS-PAGE and electrophoretically transferred onto a polyvinylidene difluoride membrane. The membranes were blocked with TBS containing 4% skim milk for 30 min. After rinsing in TBS containing 0.1% Tween 20, the membranes were incubated with a primary antibody solution for 1 h at room temperature or overnight at 4 °C, followed by incubation with HRP-conjugated secondary antibodies. They were rinsed again and exposed to EzWestLumi One (WSE-7110, ATTO, Tokyo, Japan). After rinsing with 10% H_2_O_2_ to inactivate HRP, each membrane was hybridized with HRP-conjugated anti-beta actin antibody as loading controls.

### 2.6. Fluorescence Immunohistochemistry

Cells were grown on coverslips coated by Cellmatrix Type I-A (KP-2020, Nitta gelatin, Osaka, Japan). The samples were fixed in 1% paraformaldehyde and 0.1% Triton-X for 10 min at room temperature. After being washed with PBS, they were preincubated in PBS containing 5% skim milk. The cells were subsequently incubated overnight at 4 °C with primary antibodies in PBS, then rinsed again with PBS, followed by a reaction for 1 h at room temperature with appropriate secondary antibodies. All samples were examined using a laser-scanning confocal microscope (FV1000 v.4.02, Olympus). Photographs were processed with Photoshop CC v.22.2.0 (Adobe).

### 2.7. Flow Cytometry

ADSCs were resuspended in ice-cold Hank’s Balanced Salt Solution (HBSS) and were stained for 30 min on ice with a monoclonal antibody. Propidium iodide (PI; 2 µg/mL) was used to eliminate dead cells from the flow cytometric analysis. The cell population in which doublet cells were gated out from the PI-negative population was used as FACS data. Flow cytometric analysis were performed on FACSAria (Becton Dickinson, Franklin Lakes, NJ, USA), and the data were analyzed using the Flowjo software (Becton Dickinson, Franklin Lakes, NJ, USA).

### 2.8. Cell Adhesion Assay

For the cell adhesion assay, 20, 100 or 500 ng of rJAM-B, rJAM-C, normal mouse IgG and Cellmatrix type I-A collagen (KP-2020, Nitta gelatin, Osaka, Japan) were plated on 96-well plates and incubated overnight at 4 °C. Wells were blocked with 10% bovine serum albumin for 2 h at 37 °C and washed with PBS. The cultured ADSCs were then stripped by 0.25% trypsin–EDTA solution. After centrifugation and resuspension by DMEM with 10% FBS, 10,000 cells were placed on each well with 100 µL of culture medium. They were incubated for 30 min at 37 °C in a CO_2_ incubator. Each well was then stained by 1% crystal violet solution. After washing 10 times with PBS, the attached cells were dissolved by 5% SDS. Finally, the absorbance of each well was measured at 560 nm.

### 2.9. Cell Proliferation Assay

The cell proliferation index was evaluated by the incorporation of bromodeoxyuridine (5-bromo-2-deoxyuridine; BrdU; B5002, Sigma-Aldrich, St. Louis, MO, USA). Twenty-four hours after passaging, ADSCs were exposed to 10 µM of BrdU for 2 h. The specimens were fixed with 4% paraformaldehyde and 0.1% Triton-X, followed by immunostaining with anti-BrdU antibody.

### 2.10. Statistical Analysis

Statistical significance for cell proliferation, cell adhesion and quantitative PCR analyses was analyzed using the Welch’s *t*-test. *p*-values < 0.05 were considered to indicate a significant result. All statistical analysis were performed by SPSS Statistics v.26 (IBM). All the data shown in figures are the representative of more than two and mostly three independent experiments that showed similar results.

## 3. Results

### 3.1. JAM-B and JAM-C Are Expressed on Cell Surfaces in Mouse ADSCs

We first determined which JAM subtypes were expressed in mouse ADSCs. Among the members of the JAM family, *Jam-A*, *Jam-B*, *Jam-C*, *Car* and *Clmp* transcripts, but not *Jam-4*, *Jam-L* or *Esam* transcripts, were detected in ADSCs by RT-PCR analysis (Figure 1A). Western blot analysis revealed that JAM-B, JAM-C and CLMP proteins were expressed in cultured ADSCs, though the expression levels of CLMP were decreased at passage 8 (Figure 1B). In contrast, JAM-A and CAR proteins were hardly detected. On immunofluorescent analysis, JAM-B and JAM-C seemed to be concentrated along the cell membranes of ADSCs, whereas CLMP was diffusely observed in the whole cytoplasm (Figure 1C). FACS analysis showed that JAM-B and JAM-C were detected at about 30% of the SVF in the mouse adipose tissue (Figure 2A). Among four representative positive markers for mouse MSCs [24], the expression levels of CD44 and Sca-1 were correlated to the JAM-B expression, and those of CD105, CD140a and Sca-1 were associated with the JAM-C expression. In addition, JAM-C was expressed in nearly 100% of the Sca1^+^/CD140a^+^/CD31^−^/CD45^−^/Ter119^−^ lineage among the SVF derived from mouse adipose tissues (Figure 2B), indicating that JAM-C represents a novel cell surface marker for MSCs.

### 3.2. JAM-C Is Broadly Observed in Mouse Adipose Interstitial Tissues

We next assessed by immunohistochemistry the distribution of JAM-B and JAM-C in adult mouse adipose tissues, and especially focused on that in the SVF. JAM-B and JAM-C were at least in part colocalized in interstitial tissues, and JAM-C appeared to be broadly observed in the interstitium compared with JAM-B (Figure 3A). JAM-C was not only concentrated with the endothelial markers ZO-1 and claudin-5 (CLDN5), but also diffusely distributed around microvessels in the interstitium (Figure 3B). In addition, type III collagen, the extracellular matrix (ECM) marker for adipose tissues [25], was at least in part colocalized with JAM-C but hardly codistributed with JAM-B (Figure 3C). A similar pattern was also observed for JAM-C and another ECM marker, heparan sulfate proteoglycan. Taken together, these observations suggest that JAM-C is extensively distributed in the ECM of the fat interstitial tissues.

### 3.3. JAM-C Is Accumulated in the Adipose Tissue as a Soluble Form

The extracellular domains of three JAM proteins, including JAM-C, are known to be cleaved and secreted as soluble forms [26,27,28]. Therefore, we subsequently verified whether the JAM-C ectodomain was shed in both ADSCs and mouse tissues. For this purpose, we used an antibody against the N-terminal ectodomain of JAM-C (JAM-C (N) Ab) that recognizes not only full-length JAM-C (fJAM-C) but also soluble JAM-C (sJAM-C) (Figure 4A). On Western blot analysis using the JAM-C (N) Ab, fJAM-C (38 kDa) and sJAM-C (28 kDa) were detected in whole-cell extracts and culture medium supernatants of ADSCs, respectively, and their signal intensity was extensively decreased by knockdown of the *Jam3* gene encoding JAM-C (Figure 4A–C). In contrast, sJAM-B was not observed in the supernatant of the cultured ADSCs by Western blot using JAM-B (N) Ab. sJAM-C but not sJAM-B was also evident in adult mouse spleen and adipose tissues (Figure 4D). Moreover, by double immunofluorescence analysis, the JAM-C (N)-immunoreactive signal was diffusely observed in the interstitial spaces of adult mouse adipose tissues, whereas the JAM-C (C) signal was mainly restricted to the cytoplasm and cell membranes of small round cells in the interstitium (Figure 4E). Thus, JAM-C was cleaved and secreted as a soluble form in vitro and in vivo, suggesting that sJAM-C could be deposited in extracellular spaces of fat interstitial tissues.

### 3.4. sJAM-C Acts as the Niche for ADSCs

We then evaluated whether the extracellular sJAM-C exhibited the scaffolding function of mouse ADSCs. When ADSCs were grown in culture plates coated with recombinant JAM-C (rJAM-C) protein, which corresponds to the extracellular domain of JAM-C, the cell adhesion ability was significantly and dose-dependently increased compared with the negative control IgG (Figure 5A,B). On the other hand, rJAM-B enhanced ADSC adhesion only at the highest dose tested and less than rJAM-C and type I collagen. In addition, cell proliferation of ADSCs was significantly increased not by rJAM-B coating, but by rJAM-C overlay (Figure 5C). Moreover, on RT-PCR analysis, mRNA expression levels of five MSC markers, *Cd44*, *Cd105*, *Cd140a*, *Cd166* and *Sca-1*, in the mouse ADSCs were significantly elevated by coating with rJAM-C compared to that of IgG (Figure 5D). rJAM-B increased *Cd44* and *Cd105* mRNA amounts less efficiently than rJAM-C, but did not alter the *Cd140a*, *Cd166* or *Sca-1* transcript amounts. Hence, sJAM-C in the extracellular matrix appeared to play a scaffolding role and participated not only in promoting cell growth but also in maintaining ADSCs as a niche-like microenvironment.

### 3.5. The sJAM-C/JAM-B Complex Contributes to Maintaining ADSCs

Since fJAM-C and sJAM-C bind to fJAM-B stronger than fJAM-C [16,29], we subsequently determined whether sJAM-C formed a complex with fJAM-B on cultured ADSCs. Immunoprecipitation assay revealed that sJAM-C, but not fJAM-C, was associated with JAM-B on ADSCs (Figure 6). We further validated whether the biological function of sJAM-C was mediated via JAM-B on ADSCs. To this end, the expression of the mouse *Jam2* gene encoding JAM-B in ADSCs was suppressed using CRISPR/Cas9-based genome editing (Figure 7A). Prominent knockdown of JAM-B protein expression was confirmed by Western blot (Figure 7B). The rJAM-C-enhanced cell adhesion was significantly decreased by suppression of JAM-B expression in ADSCs (Figure 7C). Furthermore, the rJAM-C-induced gene expression of *Cd44*, *Cd105*, *Cd140a* and *Cd166* was significantly reversed by JAM-B knockdown in ADSCs (Figure 7D). Thus, the sJAM-C/fJAM-B interaction contributes to ADSC adhesion and maintenance.

## 4. Discussion

In the present study, we demonstrated, using RT-PCR and Western blot analyses, that JAM-B and JAM-C were expressed in mouse ADSCs. Our immunofluorescence and FACS analyses revealed that both JAM-B and JAM-C proteins were distributed on cell surfaces of ADSCs. Since the expression of multiple MSC markers in the SVF of mouse adipose tissue was associated with JAM-B or JAM-C expression, these JAMs may be useful to isolate and enrich ADSCs from the SVF. The Sca1^+^/CD140a^+^/CD31^−^/CD45^−^/Ter119^−^ lineage of the SVF almost exclusively showed JAM-C expression, revealing that JAM-C appears to be appropriate cell surface marker for ADSCs. On the other hand, we previously showed that both JAM-B and JAM-C were assembled at primordial cell junctions in mouse F9 stem cells [30]. Taken together with the results of previous reports [14,16,17], our results reinforce the notion that JAM-B and/or JAM-C are expressed in a variety of stem and progenitor cell types.

The second conclusion of our work was that the JAM-C ectodomain was cleaved and secreted as a soluble form in vitro and in vivo, leading to accumulation in extracellular spaces of fat interstitial tissues. This conclusion was drawn from the following results: (1) sJAM-C was detected in a supernatant of ADSCs and mouse adipose tissues by Western blot using the JAM-C (N) Ab; (2) the sJAM-C levels in a supernatant of ADSCs were markedly decreased when the JAM-C expression was suppressed by the CRISPR technique; (3) diffuse distribution of JAM-C around microvessels in the interstitium of adult mouse adipose tissues was observed by immunofluorescence analysis using the JAM-C (N) Ab but not the JAM-C (C) Ab; 4) the two extracellular matrix markers for adipose tissues, namely type III collagen and heparan sulfate proteoglycan, were colocalized with JAM-C but not with JAM-B in the fat. Extracellular deposition of JAM-C has also been reported in the subepithelial limbal matrix, though the accumulation of either fJAM-C or sJAM-C was not distinguished [31]. Because both adipose and limbal tissues are rich in reticular fibers consisting of type III collagen [25,32], it is worth noting that JAM-C is accumulated in the extracellular matrix of these two tissues. In addition, since type III collagen-based reticular fibers are also abundant in bone marrow, lymph node and spleen [25], sJAM-C might be deposited in interstitial spaces of these organs. In fact, our Western blot analysis revealed that sJAM-C was detected in mouse spleen tissue.

We also showed that rJAM-C protein corresponding to sJAM-C promoted ADSC adhesion on a culture dish. Importantly, when ADSCs were grown in culture plates coated with rJAM-C protein, the expression of five distinct MSC markers and cell renewal were significantly enhanced, indicating that sJAM-C accumulated in the extracellular matrix possesses a scaffolding function and participates in maintaining ADSCs. Thus, a novel culture system of ADSCs using rJAM-C as a biomaterial might be a valuable tool to mimic the interaction between MSCs and the niche in vivo. sJAM-C may also act as the niche-like microenvironment for a variety of stem cells if it is deposited in interstitial spaces of not only fat but also other organs, as described above. Moreover, the effects of JAM-C on various physiological and pathological processes, including the self-renewal of leukemia-initiating cells and malignant behavior of melanoma and lung adenocarcinoma cells [16,33,34,35], might be at least in part attributed to the shedding of the JAM-C ectodomain and its accumulation in the extracellular matrix.

It is known that fJAM-C and sJAM-C bind with higher affinity to fJAM-B than fJAM-C [16,29,36,37,38]. In the current study, we demonstrated by immunoprecipitation assay that sJAM-C formed a complex with fJAM-B on ADSCs. In addition, our knockdown study revealed that the heterophilic sJAM-C/fJAM-B interaction was involved in ADSC adhesion and maintenance. Thus, the sJAM-C/fJAM-B complex represents the functional unit transducing the sJAM-C signal, though the sJAM-C/fJAM-C interaction may be at least in part involved in the signaling (Figure 8). It is unknown how the sJAM-C/fJAM-B signaling induces the gene expression of several MSC markers and cell proliferation in ADSCs. However, we have recently identified that the cell adhesion signal initiated by another tight-junction protein, claudin-6, regulates the activity of nuclear receptors such as the retinoic acid receptor γ and estrogen receptor α [39]. Therefore, similar to claudin-6 signaling, sJAM-C/fJAM-B signaling may link to certain transcription factors.

## 5. Conclusions

In conclusion, we found that the JAM-C ectodomain is shed and sJAM-C was broadly accumulated in interstitial spaces of the adipose tissue. We also identified that the sJAM-C is coupled with fJAM-B to stimulate the cell adhesion, proliferation and maintenance of ADSCs. These findings provide insight into potential biological activity of the cleaved ectodomain of JAM-C as the scaffolding matrix protein and the niche-like microenvironment for ADSCs. The extracellular domains of other cell–cell adhesion molecules may be also excised from normal and cancer stem cells and deposited in the interstitium as bioactive soluble forms to generate an adequate niche-like microenvironment that is beneficial to their own maintenance.

## Figures and Tables

**Figure 1 biomedicines-09-00278-f001:**
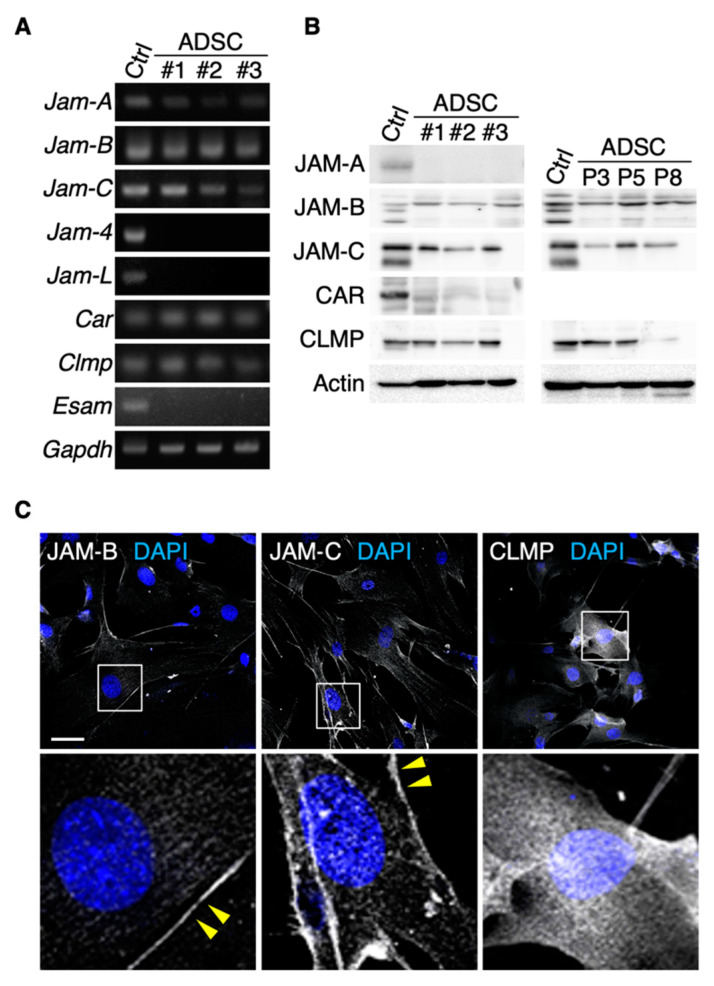
JAM-B and JAM-C are concentrated on cell membranes of mouse adipose-derived stromal/stem cells (ADSCs). RT-PCR (**A**) and Western blot (**B**) for the indicated molecules in ADSCs derived from three mice (#1–3). Mouse kidney (for *Jam-4*) and spleen tissues (for other junctional adhesion molecules; JAMs) are used as positive controls (Ctrl). P, passage. (**C**) Confocal images of ADSCs stained for the indicated markers. Arrowheads show JAM-B- and JAM-C-immunoreactive signals on the cell membranes of ADSCs. Squares indicate the enlarged areas. Scale bar, 50 µm.

**Figure 2 biomedicines-09-00278-f002:**
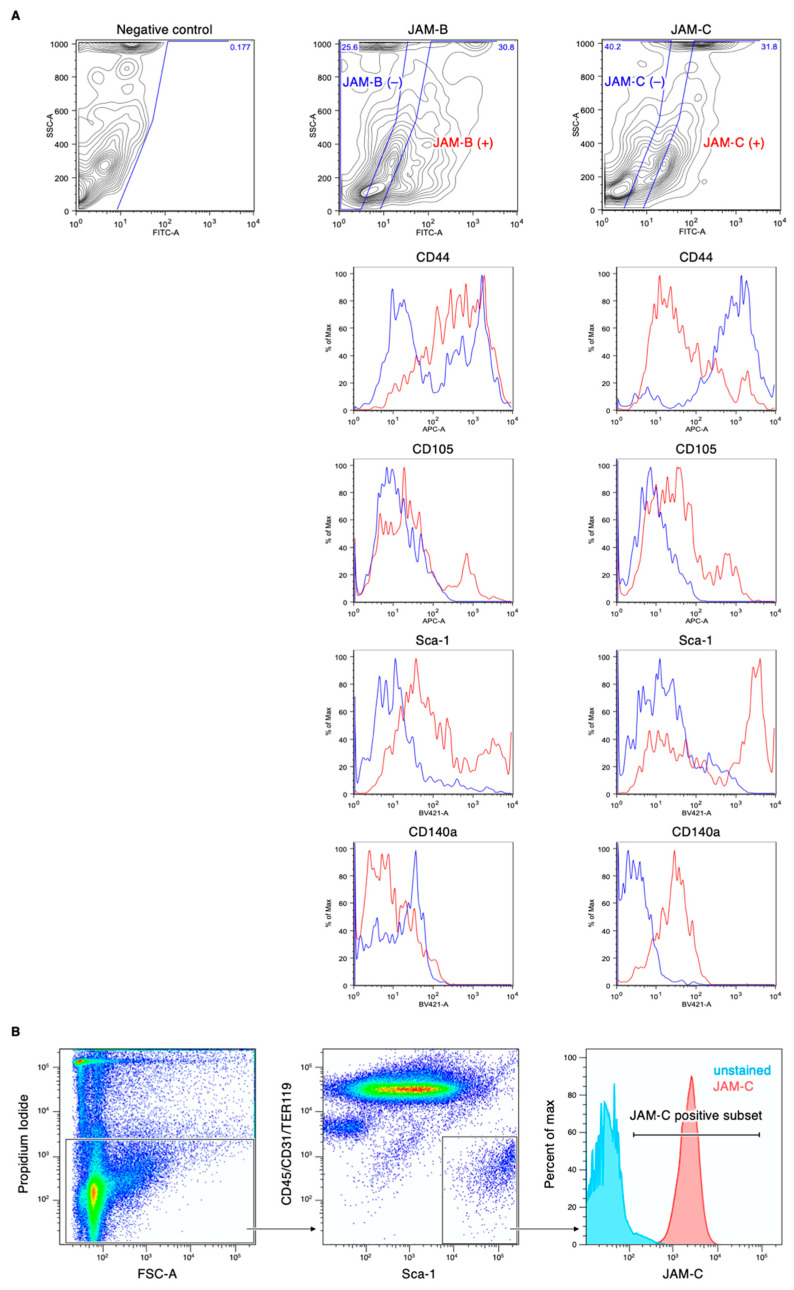
Fluorescence-activated cell sorting (FACS) profiles of the stromal vascular fraction (SVF) of the mouse adipose tissue. (**A**) Association between the expression of mesenchymal stem cell (MSC) markers CD44, CD105, CD140a and Sca-1 and JAM-B or JAM-C expression. (**B**) The JAM-C expression in the Sca1^ ^/CD31^−^/CD45^−^/Ter119^−^ cell lineage.

**Figure 3 biomedicines-09-00278-f003:**
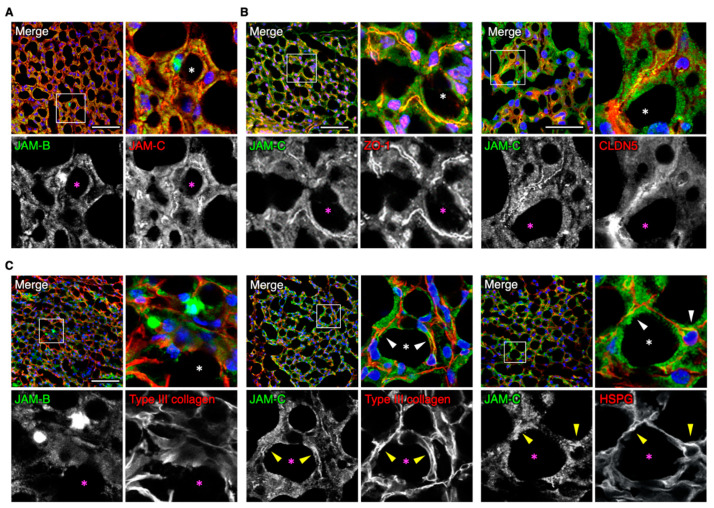
JAM-C is widely distributed in the fat interstitial tissues. (**A**–**C**) Confocal images of mouse ADSCs stained for the indicated markers. Asterisks indicate lipid droplets in differentiated adipocytes, and arrowheads show colocalization of JAM-C and ether type III collagen or heparan sulfate proteoglycan (HSPG). Squares show the enlarged areas. Scale bars, 50 µm.

**Figure 4 biomedicines-09-00278-f004:**
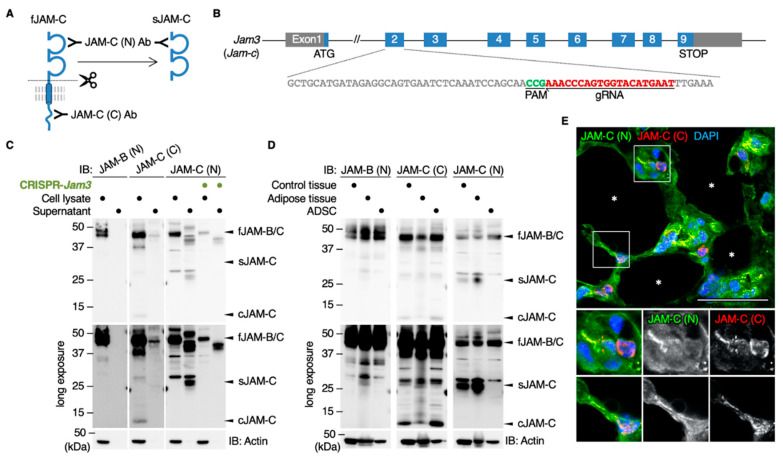
The cleaved JAM-C ectodomain is accumulated in the fat interstitial tissues. (**A**) Schematic illustration for detecting full-length JAM-C (fJAM-C) and/or soluble JAM-C (sJAM-C) using JAM-C (N) and JAM-C (C) antibodies (Abs) targeting the N- and C-termini, respectively. (**B**) Knockdown of the Jam3 gene encoding mouse JAM-C in ADSCs using the CRISPR method. (**C**) Western blot for the indicated proteins in the whole-cell lysates and supernatants of the revealed ADSCs. (**D**) Western blot for the indicated proteins in the whole-cell lysates of mouse adult spleen tissue (control), adipose tissue and ADSCs. (**E**) Confocal images of mouse adipose tissue stained with JAM-C (N) and JAM-C (C) Abs. Asterisks indicate lipid droplets in differentiated adipocytes. Squares show the enlarged areas. Scale bar, 50 µm.

**Figure 5 biomedicines-09-00278-f005:**
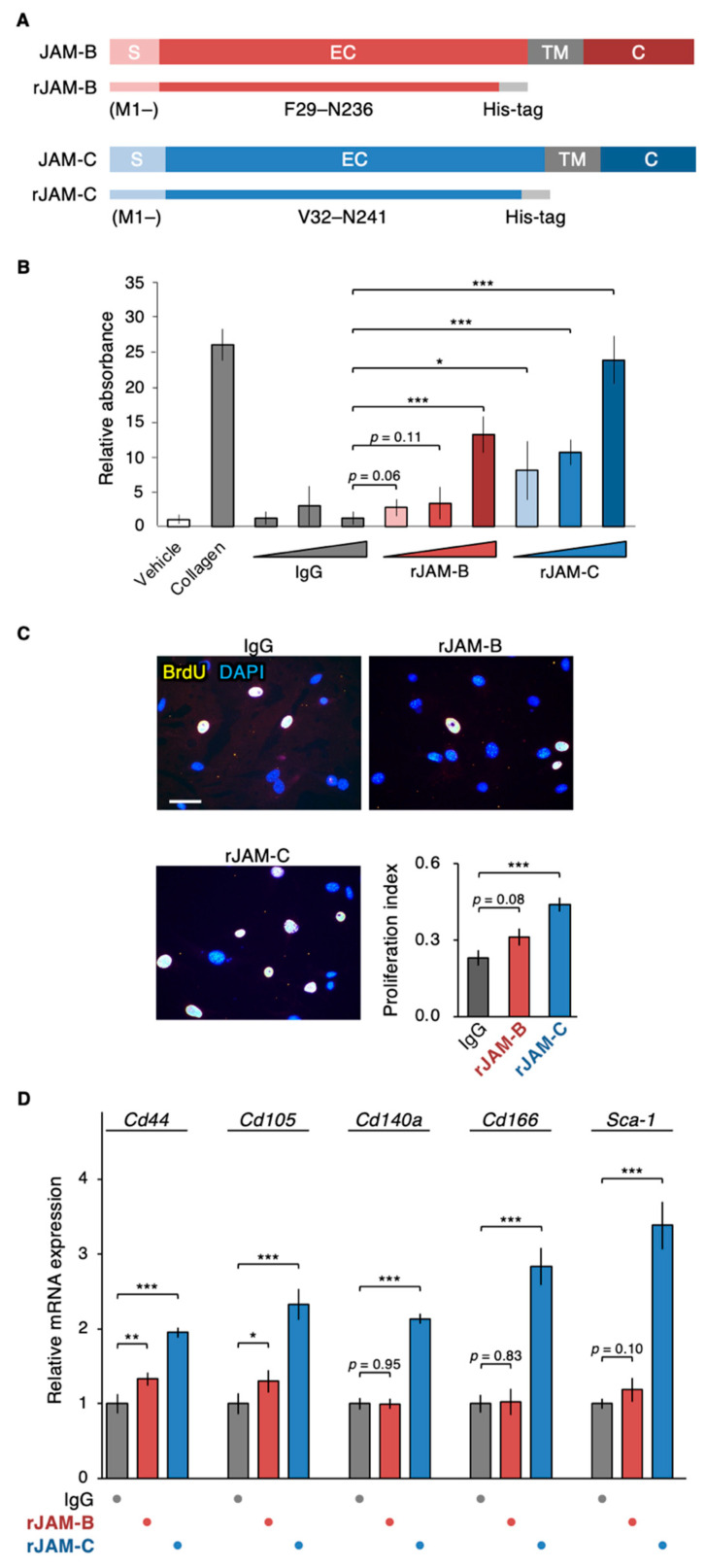
Soluble JAM-C functions as the niche for ADSCs. (**A**) Schematic illustration of recombinant JAM-B (rJAM-B) and rJAM-C. S, signal domain; EC, extracellular domain; TM, transmembrane domain; C, cytoplasmic domain. (**B**) Cell adhesion assay for ADSCs grown on culture dishes coated with the indicated proteins. The relative levels are shown in histograms (mean ± SD; *n* = 5). (**C**) BrdU assay for ADSCs grown on culture plates coated with the indicated proteins. The BrdU/DAPI levels are shown in histograms (mean ± SD; *n* = 12). Scale bar, 100 µm. (**D**) RT-qPCR for the indicated MSC markers in ADSCs grown on culture dishes coated with the indicated proteins. The relative expression levels are shown in the histograms (mean ± SD; *n* = 4). * *p* < 0.05; ** *p* < 0.01; *** *p* < 0.001.

**Figure 6 biomedicines-09-00278-f006:**
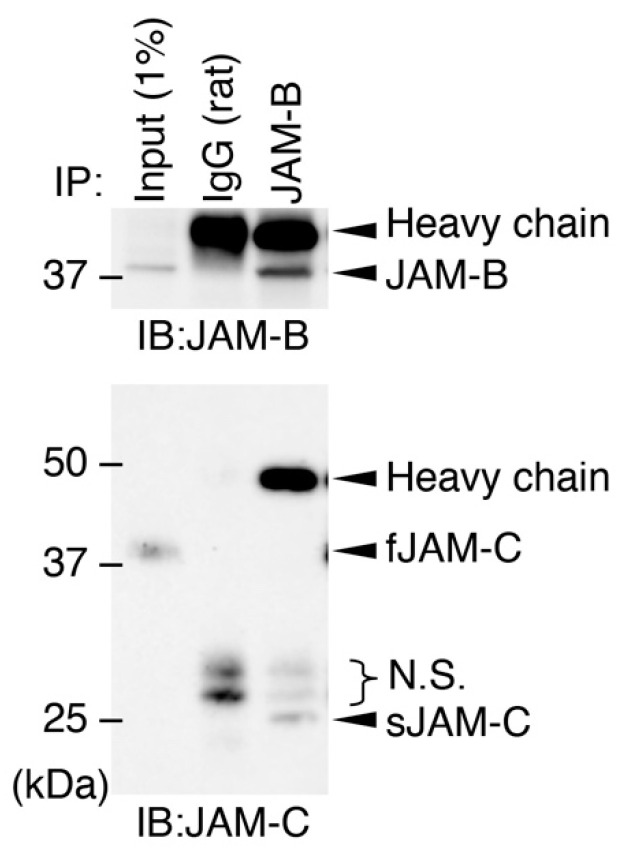
Association between soluble JAM-C and full-length JAM-B on ADSCs. Whole-cell lysates (1% of the input protein samples) and the samples immunoprecipitated (IP) with IgG or JAM-B Ab were immunoblotted (IB) with the indicated Abs. N.S., nonspecific signals.

**Figure 7 biomedicines-09-00278-f007:**
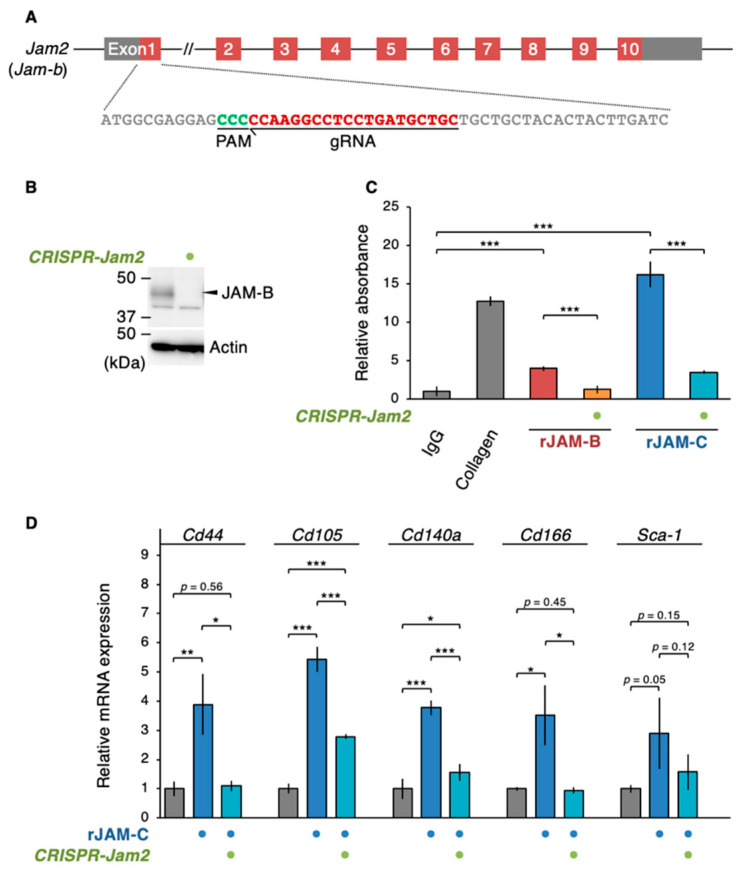
A Soluble JAM-C couples with JAM-B to promote the ADSC adhesion and niche function. (**A**) Knockdown of the *Jam2* gene encoding mouse JAM-B in ADSCs using the CRISPR method. (**B**) Western blot for the indicated proteins in the whole-cell lysates of the revealed ADSCs. (**C**) Cell adhesion assay for ADSCs grown in the indicated culture conditions. The relative levels are shown in histograms (mean ± SD; *n* = 8). (**D**) RT-qPCR for the indicated MSC markers in ADSCs cultivated in the indicated culture conditions. The relative expression levels are shown in the histograms (mean ± SD; *n* = 4). * *p* < 0.05; ** *p* < 0.01; *** *p* < 0.001.

**Figure 8 biomedicines-09-00278-f008:**
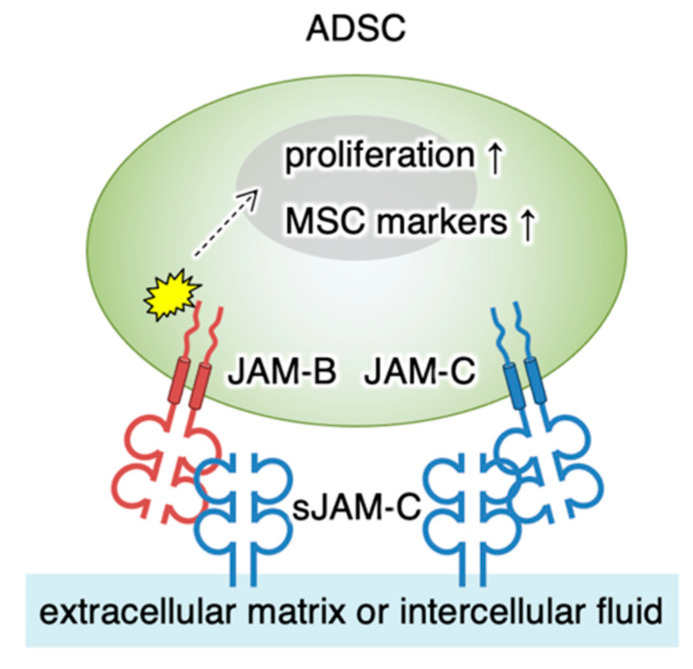
Schematic model for the regulation of ADSC functions by soluble JAM-C. The soluble JAM-C (sJAM-C) deposited on the extracellular matrix interacts with JAM-B and partly with JAM-C on ADSC. sJAM-C/JAM-B signaling reaches the nucleus (dashed arrow) and stimulates cellular proliferation and expression of MSC markers (arrows). MSC, mesenchymal stromal/stem cell.

**Table 1 biomedicines-09-00278-t001:** Antibodies.

Antibody	Type	Host	IHC	IB	IP	FACS	Source
JAM-A	pAb	Rabbit		1:1000			Thermo Fisher Scientific (36-1700)
JAM-B	mAb	Rat	1:100	1:2000	2 µg		R&D Systems (MAB988)
JAM-C (N)	pAb	Goat	1:200	1:1000			R&D Systems (AF1213)
JAM-C (C)	pAb	Rabbit	1:200	1:1000			Thermo Fisher Scientific (40-9000)
CAR	pAb	Rabbit	1:50	1:500			Bioss (bs-2389R)
CLMP	pAb	Rabbit	1:100	1:1000			Gene Tex (GTX51678)
Claudin-5	mAb	Mouse	1:300				Thermo Fisher Scientific (35-2500)
ZO-1	pAb	Rabbit	1:100				Thermo Fisher Scientific (61-7300)
Type III collagen	pAb	Rabbit	1:100				abcam (ab7778)
HSPG	mAb	Rat	1:300				Merck Millipore (MAB1948P)
CD31	mAb	Rat				1:200	BD Pharmingen (550274)
CD44	mAb	Rat				1:200	Biolegend (103011)
CD45	mAb	Rat				1:200	BD Pharmingen (550539)
CD105 (ENG)	mAb	Rat				1:200	Biolegend (120413)
CD140a (PDGFRa)	mAb	Rat				1:200	BD Pharmingen (562774)
Sca-1 (LY6a)	mAb	Rat				1:200	BD Pharmingen (562729)
TER119	mAb	Rat				1:200	Thermo Fisher Scientific (12-591-82)
BrdU	mAb	Rat	1:500				abcam (ab220074)
β-Actin	mAb	Mouse		1:1000			Thermo Fisher Scientific (A2228)
Goat IgG (HRP)	pAb	Rabbit		1:2500			Dako (P0449)
Mouse IgG (HRP)	pAb	Sheep		1:10,000			GE Healthcare (NA931)
Rabbit IgG (HRP)	pAb	Donkey		1:5000			GE Healthcare (NA934)
Rat IgG (HRP)	pAb	Goat		1:2500			GE Healthcare (NA935)
Goat IgG (Alexa Fluor 488)	pAb	Donkey	1:200				Thermo Fisher Scientific (A11055)
Mouse IgG (Alexa Fluor 488)	pAb	Donkey	1:200				Thermo Fisher Scientific (A21202)
Rabbit IgG (Alexa Fluor 488)	pAb	Donkey	1:200				Thermo Fisher Scientific (A21206)
Rat IgG (Alexa Fluor 488)	pAb	Donkey	1:200				Thermo Fisher Scientific (A21208)
Goat IgG (Cy3)	pAb	Donkey	1:200				Jackson Immunoresearch (705-165-147)
Mouse IgG (Cy3)	pAb	Donkey	1:200				Jackson Immunoresearch (705-165-151)
Rabbit IgG (Cy3)	pAb	Donkey	1:200				Jackson Immunoresearch (705-165-152)
Rat IgG (Cy3)	pAb	Donkey	1:200				Jackson Immunoresearch (705-165-150)

Thermo Fisher Scientific, Waltham, MA, USA; R&D Systems, Minneapolis, MN, USA; Bioss, Woburn, MA, USA; Gene Tex, Irvine, CA, USA; abcam, Cambridge, United Kingdom; Merck Millipore, Burlington, MA, USA; Dako, Hovedstaden, Denmark; GE Healthcare, Chicago, IL, USA; Jackson Immunoresearch, West Grove, PA, USA.

**Table 2 biomedicines-09-00278-t002:** Primers for RT-PCR.

Gene	Forward Primer	Reverse Primer	Product Size (bp)
*JAM-A (F11r)*	AGCCAGATCACAGCTCCCTA	CATTGTCCTTCCGGGTCACA	672
*JAM-B (Jam2)*	TGGTCAATACCTGTGAAACACAAA	TGGACAACTAATTGCTAAAAGGG	161
*JAM-C (Jam3)*	TGTGCAAGTGAAGCCAGTGA	AGTGGCACATCATTGCGGTA	138
*JAM4 (Igsf5)*	CCTTCCAGAAAAACGCAGCA	GTCTCCCGGGTGATTCCAAA	182
*JAML (Amica1)*	GATCGCGGTGGACTGTTCTT	GCCGTCCTTGACTCACTCTAC	351
*CAR (Cxadr)*	AACGATGTCAAGTCTGGCGA	TTCCGATCCATCCACGAAGC	171
*CLMP (Clmp)*	CCTCTTTCTCCAGTCGGTTTTC	GGTTAGGGAGGAGAAGGCGA	90
*ESAM (Esam)*	AGACACCGTGTGTCCAACTC	AGTCCCAGGAACAAAACCCG	108
*Cd44*	AATGGCTCATCATCTTGGCA	GCTCACTGGGTTTCCTGTCT	150
*Cd105 (Eng)*	CAGCCAAAGTGTGGCAATCAGG	GCTACTCAGGACAAGATGGTCG	144
*Cd140a (Pdgfra)*	AGAGACTGAGCGCTGACAGT	GATGGTCTCGTCCTCTCTCT	173
*Cd166 (Alcam)*	AGGAACATGGCGGCTTCAACGA	ACACCACAGTCGCGTTCCTACT	142
*Sca-1 (Ly6a)*	CCTACCCTGATGGAGTCTGTGT	CACGTTGACCTTAGTACCCAGG	143
*Gapdh*	ATGTGTCCGTCGTGGATCTGA	TTGAAGTCGCAGGAGACAACCT	145

## Data Availability

The FACS data presented in this study are openly available at https://www.fmu.ac.jp/home/p2/Yamazaki_Biomedicines_2021.zip, accessed on 9 March 2021. Other data presented in this study are available on request from the corresponding author.

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
