# Peer review of "Soluble JAM-C Ectodomain Serves as the Niche for Adipose-Derived Stromal/Stem Cells"

_biomedicines, 2021, doi:10.3390/biomedicines9030278_

Round 1
Reviewer 1 Report
The authors demonstrated that JAM-C expressed in adipose-derived stem cells (ADSCs) maintain ADSC function by acting as the niche. The experimental designs were well polished. The approach and the finding are very interesting, but this reviewer has several comments as described below:
Major comments
#1. L54-71, the authors have introduced general information of JAMs, but it is better to describe details of role of JAMs in stem cell biology or previous researches in this field than general information.
#2. Detail information about rJAM-B and -C are needed. The authors should mention why IgG is available as negative control and the protein concentration of cell adhesion assay.
#3. Fig1D looks like showing main cell group in upper area of the plot, the authors should explain about the region of ADSCs and the gate for analysis.
#4. Fig.2c, this reviewer cannot agree the interpretation of image data.
#5. Fig3e, this reviewer cannot agree the interpretation of image data. In order to test the co-localization of sJMA-C with ECM, other analysis as such as confocal microscopy, immunohistochemistry, cultured cell analysis, or EIA assay are needed. Or the authors can revise the interpretations according to the data.
#6. Appropriate statistical methods should be used.
#7. Fig. 7, the data did not support this model. Moreover, the authors should indicate the evidences that of the binding of JAM-C and ECM and the possibility of JAM-C mediated cell-cell interactions.
#8. In discussion, the authors should discuss about interaction or affinity of s/fJAM-C to JAM-C, JAM-B, and ECMs with referring previous literatures.
#9. The authors should mention about number of experiments and reproducibility in all experiments.
Minor comments
#1. L49-L52, references are needed. Do endothelia mean endothelium or endothelial cells?
#2. L69, could you check highlight?
#3. Fig2b, more clear JAM-C image (right panel) which is well adjusted is better.
#4. Fig. 3, M.W. of s/fJAM-C should be indicated.
#5. Fig.4c and 6d, how about cell surface protein expression?
Author Response
Responses to the comments of Reviewer #1
We are grateful to the reviewer’s comments “The experimental designs were well polished. The approach and the finding are very interesting”.
We also greatly appreciate the constructive critiques that have helped to improve our manuscript.
Major comments
- L54-71, the authors have introduced general information of JAMs, but it is better to describe details of role of JAMs in stem cell biology or previous researches in this field than general information.
We describe previous findings on the contribution of JAMs to stem cell biology in lines 73-80. In fact, there are very few studies concerning their functional significance in stem cells.
- Detail information about rJAM-B and -C are needed. The authors should mention why IgG is available as negative control and the protein concentration of cell adhesion assay.
We included schematic illustration of rJAM-B/C in Fig. 5A. In addition, we mentioned why IgG was used as negative control (lines 109-111). Other information is written in lines 106-108 and line 175.
- Fig1D looks like showing main cell group in upper area of the plot, the authors should explain about the region of ADSCs and the gate for analysis.
The cell population of the upper area of the plot appears to be cell aggregates. We mentioned “The cell population in which doublet cells were gated out from the PI-negative population was used as FACS data.” in the Materials and Methods section (lines 170-172).
- 2c, this reviewer cannot agree the interpretation of image data.
We corrected the description to “type III collagen, the extracellular matrix (ECM) marker for adipose tissues, was at least in part colocalized with JAM-C but hardly codistributed with JAM-B” (lines 231-233). In addition, we added arrowheads in the Fig. 3C (former Fig. 2C) to show colocalization of JAM-C and either Type III collagen or HSPG. Please check the accompanied Figure file, which provides a better resolution of images.
- Fig3e, this reviewer cannot agree the interpretation of image data. In order to test the co-localization of sJAM-C with ECM, other analysis as such as confocal microscopy, immunohistochemistry, cultured cell analysis, or EIA assay are needed. Or the authors can revise the interpretations according to the data.
According to review’s comment, we revised the interpretation to “Thus, JAM-C was cleaved and secreted as a soluble form in vitro and in vivo, suggesting that sJAM-C could be deposited in extracellular spaces of fat interstitial tissues” (lines 256-258).
- Appropriate statistical methods should be used.
Although all the quantitative data presented in this manuscript theoretically distribute in equal variances, to cover the possibility of unequal variances, Welch’s t-test was chosen. But the authors also verified the results using Mann–Whitney U test, a nonparametric test, and no significant discrepancies were detected.
- 7, the data did not support this model. Moreover, the authors should indicate the evidences that of the binding of JAM-C and ECM and the possibility of JAM-C mediated cell-cell interactions.
To meet the criticism, we corrected Figure 8 (former Figure 7) as much as possible.
- In discussion, the authors should discuss about interaction or affinity of s/fJAM-C to JAM-C, JAM-B, and ECMs with referring previous literatures.
We discussed about affinity of fJAM-C and sJAM-C to JAM-B and JAM-C (lines 362-364).
- The authors should mention about number of experiments and reproducibility in all experiments.
All the data shown in the figures are representative of more than two, mainly three independent experiments, which showed similar results. The authors added the description in the materials and methods section (line 191).
Minor comments
- L49-L52, references are needed. Do endothelia mean endothelium or endothelial cells?
We included an appropriate reference (lines 49-52) and corrected endothelia to endothelial cells (line 51).
- L69, could you check highlight?
We deleted the highlight.
- Fig2b, more clear JAM-C image (right panel) which is well adjusted is better.
To meet the concern, we showed more clear JAM-C images that were well adjusted.
- 3, M.W. of s/fJAM-C should be indicated.
We indicate M.W in the text (lines 247 and 248).
- 4c and 6d, how about cell surface protein expression?
We do not have data on expression of cell surface protein. Since it is pretty difficult to perform an additional experiment within 7 days, we would like to determine it in future experiments.

Reviewer 2 Report
In this manuscript, Yamazaki et al. showed that soluble JAM-C is upregulated mesenchymal stem marker expression in adipose-derived stromal/stem cells (ADSCs) via JAM-B. Different to JAM-B, JAM-C is often cleaved an extracellular domain of JAM-C in adipose tissue, produced soluble form and soluble JAM-C accumulates at the extracellular matrix. Based on this JAM-C soluble form, ADSCs were increased cell proliferation and mesenchymal stem marker expression, thereby a series of ADSC’s function by soluble JAM-C implicated the self-renewal or maintenance of mesenchymal stem cell. These results are interesting and have novelty, but I have some critical issues for publication in this manuscript. Therefore, I require some critical data in revision.
Major comments
Title: please change to “… Adipose-Derived Stromal/Stem Cells”. I think that JAM-B expressed ADSCs are not “stem” cells. Because JAM-B positive ADSCs do not express CD140a (Supplemental figure 1). Many papers have shown that these stem cells express all mesenchymal marker.
Figure 1A: I don’t feel the necessity for this RT-PCR data. Especially, there is not the correlation of Jam-C #3, Car #1-#3 and Clmp #3 data with each protein data (Figure 1B).
Figure 1C: The authors show that JAM-B and JAM-C were concentrated along the cell membranes of ADSCs, whereas CLMP was diffusely observed in the whole cytoplasm. However, in this immunocytostaining, the authors perform with permeabilization by triton. The authors should carefully demonstrate the membrane localization of JAM-B, JAM-S and CLMP without triton using each N-terminus-recognizing antibody.
Figure 1D and supplemental figure 1: the authors should check JAM-B or JAM-C expression in Sca1+CD140a+Lineage(CD31,CD45,Ter119)- compartment of ADSCs. In addition, supplemental figure 1 should also represent negative control.
Figure 2: The authors should represent a constant image of JAM-C. I feel that 2B right and 2C images are too strong staining.
Figure 3D: In IB JAM-C (C) panel, I interest if the JAM-C C-terminal protein (approximately 10 kDa?) can detect. After JAM-C extracellular domain was cleaved, does JAM-C intracellular domain induce degradation immediately?
Figure 5: In lower panel, IgG lane hardly detect heavy chain. Perhaps the authors used rat IgG and rat JAM-B antibody for IP. The authors should describe in detail about used IgG and antibody’s host information.
Minor comments
Page 1 line 23: Is “mode” right spell? “model”?
Page 2 line 69: please delete yellow pen.
Page 4 line 160: please check this phrase “The antibodies used were…”.
Page 7 line249-250: where are asterisks?
Figure 7: I feel that you should insert the description of upregulation of cell proliferation and mesenchymal stem cell markers. On the other hand, this study was not demonstrated that soluble JAM-C interact JAM-C full length and this pathway induce cell proliferation and stem markers.
Author Response
Responses to the comments of Reviewer #2
We appreciate the reviewer’s comments “These results are interesting and have novelty”.
We are grateful to the various concerns to improve our manuscript.
Major comments
- Title: please change to “… Adipose-Derived Stromal/Stem Cells”. I think that JAM-B expressed ADSCs are not “stem” cells. Because JAM-B positive ADSCs do not express CD140a (Supplemental figure 1). Many papers have shown that these stem cells express all mesenchymal marker.
To meet the concern, we changed the Title to “… Adipose-Derived Stromal/Stem Cells”.
- Figure 1A: I don’t feel the necessity for this RT-PCR data. Especially, there is not the correlation of Jam-C #3, Car #1-#3 and Clmp #3 data with each protein data (Figure 1B).
We agree that there are some differences in the levels of gene expression of JAM members among three batches of ADSCs (#1-#3), most probably due to each culture condition. In addition, it is known that mRNA levels is not always correlated to the protein levels. Therefore, we would like to present the RT-PCR data to show which JAM subtypes are expression in mouse ADSCs.
- Figure 1C: The authors show that JAM-B and JAM-C were concentrated along the cell membranes of ADSCs, whereas CLMP was diffusely observed in the whole cytoplasm. However, in this immunocytostaining, the authors perform with permeabilization by triton. The authors should carefully demonstrate the membrane localization of JAM-B, JAM-C and CLMP without triton using each N-terminus-recognizing antibody.
We corrected “JAM-B and JAM-C were concentrated along the cell membranes” to “JAM-B and JAM-C seemed to be concentrated along the cell membranes” (lines 202-203). Since we detected JAM-B and JAM-C expression in unfixed/unpermeabilized ADSCs by FACS analysis, we conclude that both JAMs are expressed on cell surfaces of ADSCs.
- Figure 1D and supplemental figure 1: the authors should check JAM-B or JAM-C expression in Sca1+CD140a+Lineage(CD31,CD45,Ter119)-compartment of ADSCs. In addition, supplemental figure 1 should also represent negative control.
According to review’s comment, we analysed the JAM-C expression in Sca1+/CD31-/CD45-/Ter119- cells of the SVF derived from mouse adipose tissues. Since the CD140a+ cell population is identical to Sca1+ population, we used one positive marker for this analysis. As shown in novel Figure 2B, JAM-C was expressed in almost all cells of the lineage, indicating that JAM-C represents a novel cell surface marker for MSCs (lines 207-210). Unfortunately, we do not keep the data on the JAM-B expression in the same lineage, and are not able to perform an additional experiment to finish revision within 7 days as required by editors. We also added a negative control in Figure 2A (former Supplementary Figure S1).
- Figure 2: The authors should represent a constant image of JAM-C. I feel that 2B right and 2C images are too strong staining.
We carefully analysed data and showed representative images for JAM-C.
- Figure 3D: In IB JAM-C (C) panel, I interest if the JAM-C C-terminal protein (approximately 10 kDa?) can detect. After JAM-C extracellular domain was cleaved, does JAM-C intracellular domain induce degradation immediately?
The issue would be interesting. We would like to clarify the nature of JAM-C intracellular domain in future experiments.
- Figure 5: In lower panel, IgG lane hardly detect heavy chain. Perhaps the authors used rat IgG and rat JAM-B antibody for IP. The authors should describe in detail about used IgG and antibody’s host information.
We corrected “IgG” to “IgG (rat)” for IP in the novel Figure 6. In addition, we presented the detailed information on IgG and antibodies used in this study as Table 1 instead of Supplementary Table S1.
Minor comments
- Page 1 line 23: Is “mode” right spell? “model”?
We used “mode” as a meaning of “type” or “kind”.
- Page 2 line 69: please delete yellow pen.
We deleted the highlight.
- Page 4 line 160: please check this phrase “The antibodies used were…”.
We deleted the phrase.
- Page 7 line249-250: where are asterisks?
We added asterisks in the Figure.
- Figure 7: I feel that you should insert the description of upregulation of cell proliferation and mesenchymal stem cell markers. On the other hand, this study was not demonstrated that soluble JAM-C interact JAM-C full length and this pathway induce cell proliferation and stem markers.
To meet the criticism, we included “cell proliferation ↑” and “expression of mesenchymal stem cell markers ↑” and corrected this Figure.

Reviewer 3 Report
The manuscript describes an interesting study with some potentially important findings. The potential for the use of JAM-C surface expression as a marker for ADSC identification should be explored further, and the beneficial effects of soluble JAM-C on ADSC growth and function is something that could prove to be very useful. The conclusion of the authors that soluble JAM-C provides a niche for ADSCs is probably overstated, since it is not clear from the study that the functions described in these artificial systems truly reflects their physiological function in vivo.
Minor comments:
- Figure 1D is described in the figure legend to figure 1, but there is no figure 1D.
- The supplemental materials seem to be redundant to Figure 2A, Table 1 and Table 2 in the text.
- Although the overall English is fine, there are a few places where editing would be useful.
Author Response
We are grateful to the reviewer’s comments “The manuscript describes an interesting study with some potentially important findings”.
To meet the criticism, we revised the conclusion as follows.
“These findings provide insight into potential biological activity of the cleaved ectodomain of JAM-C as the scaffolding matrix protein and the niche-like microenvironment for ADSCs” (lines 380-382).
Minor comments
- Figure 1D is described in the figure legend to figure 1, but there is no figure 1D.
We deleted the figure legend for Figure 1D (line 220).
- The supplemental materials seem to be redundant to Figure 2A, Table 1 and Table 2 in the text.
We totally removed the Supplementary materials.
- Although the overall English is fine, there are a few places where editing would be useful.
We edited English again.
Reviewer 4 Report
The manuscript by Yamazaki et al. studied adipose-derived cells that include mesenchymal stem cells, showing a role for junctional adhesion molecules in adhesion, proliferation and maintenance of these cell populations. The study is carefully executed and shows important findings. My main concern involves the definitions of adipose-derived stem cells, mesenchymal stem cells and niche. Otherwise, the paper appears to be a strong contribution to the field.
1. The use of the term “stem cell” in reference to cell populations (adipose-derived stem cells and mesenchymal stem cells) is not being used rigorously, and the cell populations were not defined clearly. Stem cells are typically a minority of the overall tissue population. The stem cells divide infrequently. After an asymmetric cell division, the stem cell self-renews, and the committed cell enters the transit amplifying population. Ultimately, the derivatives terminally differentiate. The stem cell term used in the paper seems to refer to stem, transit amplifying and differentiated cell populations. Rigorous distinctions are not made. Markers used to identify “stem cells” and proliferation also are used without distinction between populations. This made the manuscript hard to interpret, and the manuscript should be edited to be much more specific.
2. The authors discuss the role of JAM-C signaling in the niche. The meaning of the niche is unclear. The authors show that the adipose-derived stem cell population expresses JAM-C. Is this autocrine signaling by the stem cells? Are the authors suggesting that the JAM-C is derived from support cells, like endothelial cell populations? This needs to be defined clearly and explicitly.
3. The term immunoblot and Western blot were both used. One or the other should be used consistently.
4. The conclusions about the niche and stem cell populations were very specific, despite the use of these terms not being very specific. The conclusions should carefully and conservatively fit the data and the definitions of these terms. For example, the niche is a very important functional entity, and in vivo experiments may be needed to test the role of JAM-C activity in the niche. Care should be taken to not overreach the data with the conclusions.
Author Response
We appreciate the reviewer’s comments “The study is carefully executed and shows important findings”.
We are grateful to the various concerns to improve our manuscript.
Specific comments
- The use of the term “stem cell” in reference to cell populations (adipose-derived stem cells and mesenchymal stem cells) is not being used rigorously, and the cell populations were not defined clearly. Stem cells are typically a minority of the overall tissue population. The stem cells divide infrequently. After an asymmetric cell division, the stem cell self-renews, and the committed cell enters the transit amplifying population. Ultimately, the derivatives terminally differentiate. The stem cell term used in the paper seems to refer to stem, transit amplifying and differentiated cell populations. Rigorous distinctions are not made. Markers used to identify “stem cells” and proliferation also are used without distinction between populations. This made the manuscript hard to interpret, and the manuscript should be edited to be much more specific.
We agree the reviewer’s criticism. To meet the concern, we carefully used the term “stem cell” throughout the manuscript. For instance, we corrected “adipose-derived stem cells” and “stem cells” to “adipose-derived stromal/stem cells” and “stem and progenitor cells”, respectively.
- The authors discuss the role of JAM-C signaling in the niche. The meaning of the niche is unclear. The authors show that the adipose-derived stem cell population expresses JAM-C. Is this autocrine signaling by the stem cells? Are the authors suggesting that the JAM-C is derived from support cells, like endothelial cell populations? This needs to be defined clearly and explicitly.
We mean that ADSCs generate a niche-like microenvironment that is beneficial to their own maintenance, via excising the ectodomain of JAM-C. The soluble JAM-C deposited on the extracellular matrix or intercellular fluid interacts with JAM-B on ADSCs, potentially propagating intracellular signaling, similar to our previous report (Sugimoto et al., PNAS, 116, 24600-24609, 2019) (lines 365-375 and 377-382).
- The term immunoblot and Western blot were both used. One or the other should be used consistently.
We consistently used the term Western blot (line 140).
- The conclusions about the niche and stem cell populations were very specific, despite the use of these terms not being very specific. The conclusions should carefully and conservatively fit the data and the definitions of these terms. For example, the niche is a very important functional entity, and in vivo experiments may be needed to test the role of JAM-C activity in the niche. Care should be taken to not overreach the data with the conclusions.
According to review’s comment, we used “niche-like environment” instead of “niche” throughout the manuscript.
Round 2
Reviewer 1 Report
The authors addressed the major issues and significantly improved the revised version of the manuscript.
I also agree with the carefully revised manuscript.
Author Response
Thanks for the reviewer’s comments “The authors addressed the major issues and significantly improved the revised version of the manuscript. I also agree with the carefully revised manuscript”.
Reviewer 2 Report
The authors have adequately revised the manuscript. I have no further comments.
Author Response
We appreciate the reviewer’s comments “The authors have adequately revised the manuscript. I have no further comments”.